# Constraint-Based Synthetic Data Generation for LLM Mathematical Reasoning

**Timofey Fedoseev**[1,2], , **Dimitar I. Dimitrov**[2,3], **Timon Gehr**[3], **Martin Vechev**[2,3]

[1] École Polytechnique      [2] INSAIT, Sofia University "St. Kliment Ohridski"      [3] ETH Zurich

{timofei.fedoseev}@polytechnique.edu [1]
{dimitar.iliev.dimitrov}@insait.ai [2]
{timon.gehr, martin.vechev}@inf.ethz.ch [3]

## Abstract

Mathematical reasoning with large language models (LLMs) is an emerging research area. A recent breakthrough is the use of off-the-shelf tools LLMs are trained to utilize to offload complex tasks they cannot perform independently. Unfortunately, this approach is limited to popular tools, as many specialized tools lack the data to train these models on. Motivated by our observation that the current tools used with LLMs are insufficient for solving counting problems, in this work, we explore the problem of using Satisfiability Modulo Theories (SMT) solvers with LLMs. Namely, we leverage the SMT grammar to generate synthetic data consisting of problem statements and their solutions represented as Python code interacting with the Z3 API. Our experiments show that fine-tuning LLMs on this dataset substantially enhances their ability to generate accurate Z3 constraint encodings and improves their overall mathematical problem-solving capabilities.

## 1   Introduction

Mathematical reasoning is a key emerging area of competence for LLMs. Instead of asking for the solution of a mathematical problem directly, a common technique is to instead ask for program code that in turn solves the problem, offloading already easily automatable parts of the mathematical reasoning to off-the-shelf libraries [1, 2, 3, 4]. This works particularly well for popular libraries (such as SymPy [3]), because large amounts of high-quality training data can be found online. Common approaches are to train LLMs on solutions obtained from stronger LLMs [5], and/or to bootstrap from a large number of samples, discarding any incorrect solutions [6]. However, not for every reasoning tool, training data is initially similarly abundant. Therefore, more elaborate data generation approaches may be helpful in order to more readily tap into the potential of less popular tools.

Motivated by the observation that existing methods based on SymPy often struggle to solve counting problems, we investigate an approach based on Z3Py [7], which excels at counting problems with a small enough answer. As expected due to the relatively lower popularity of Z3Py, even state-of-the-art LLMs such as GPT-4o are not yet proficient users of Z3Py.

In this work, we make the following contributions:

- A method for generating large-scale, high-quality synthetic data for Z3Py that can be used to enhance the mathematical problem-solving capabilities of current LLMs.
- An additional Z3Py synthetic dataset obtained through filtering of problems from existing datasets with traditional rejection sampling of solutions from LLMs.
- A quantitative evaluation on GPT-4o and GPT-4o-mini against their fine-tuned variants using the two approaches for data generation, showing that our new training data generation

38th Conference on Neural Information Processing Systems (NeurIPS 2024).

method improves the model performance on the Z3Py formalization task more than rejection sampling solutions of existing problems alone.

- A qualitative evaluation, showing the models fine-tuned on our Z3Py data can solve problems that they were unable to solve without fine-tuning, even if allowed to use arbitrary libraries.

## 2 Background: Enumerating Solutions with Z3Py

Satisfiability modulo theories (SMT) generalizes the Boolean satisfiability problem. Constraints can involve various different types of mathematical objects. An SMT solver takes in a set of constraints and determines whether the constraints can be satisfied, producing a solution ("model") if possible.

The Z3 theorem prover is a state-of-the-art open-source SMT solver. Z3Py is the Z3 API for Python. We solve counting problems with Z3Py using the technique of *blocking*: We iteratively solve constraints, adding constraints that the solution should not be one of the ones found previously. This way, the solver steps through all solutions. Once all solutions have been found, the solver returns that the constraints are unsatisfiable. This way we in particular obtain the number of possible solutions.

In some cases, state-of-the-art LLMs are able to generate Z3Py code that solves a problem specified in natural language. E.g., we prompted GPT-4o with "Please give me code using Z3Py that counts the number of ways to cover a 4x4 chessboard with 2x1 and 1x2 dominos.", and shortened the result:

```python
from z3 import *
solver = Solver()

n = 4
board = [(x, y) for x in range(n) for y in range(n)]
dom = [(Bool(f"h_{x}_{y}"), [(x, y), (x+1, y)]) for x in range(n-1) for y in range(n)] \
    + [(Bool(f"v_{x}_{y}"), [(x, y), (x, y+1)]) for x in range(n) for y in range(n-1)]
solver.add(And([Sum([v for v, pos in dom if (x, y) in pos]) == 1 for x, y in board]))

count = 0
while solver.check() == sat:
    model = solver.model()
    count += 1
    solver.add(Or([var != model[var] for var, _ in dom]))
print(count) # prints 36 (the correct answer)
```

The code first generates pairs of Z3Py Boolean variables and pairs of squares covered, for each way to place a domino. Then, it adds constraints that say that every square of the board should be covered by exactly one domino. Finally, the solutions are enumerated using the blocking technique.

Unfortunately, despite succeeding in simple cases like this one, LLMs often fail to give a correct encoding, particularly for more involved problem statements. In this work, we show how to fine-tune LLMs on the task of producing correct code to solve combinatorics problems using Z3Py.

## 3 Datasets Used for Fine-tuning

We now present our methodology for generating datasets of problems with correct Z3Py solutions.

### 3.1 Synthetic Problem Generation

To address the lack of training data for Z3py, we generate fully synthetic pairs of counting problems with corresponding Z3py solutions. We focus on four classes of ground sets: Sequences, permutations of the set $\{1, 2, \ldots, n\}$, numbers in base $k$, and subsets of the set $\{1, 2, \ldots, n\}$. For each object, we define a set of supported constraints, where each constraint includes a natural language description and corresponding Z3py code. Constraints can be applied to the object itself or to an integer parameter derived from the object, such as the sum of the sequence or the number of inversions in a permutation.

When generating a problem, we sample a ground set and a set of constraints, where each constraint has a probability of $1/2$ of being negated. These constraints are then combined to form a problem and a Z3Py solution. We run the solution and retain only the problems that finish within the time limit, returning a non-zero answer. Below is an example of a generated synthetic problem:

A subset of the set $\{1, 2, \ldots, 6\}$ (no two elements in the subset are consecutive integers) and (no three elements in the subset form an arithmetic progression) and (the subset sum is not divisible by $10$). Count the number of valid objects.

Here is the relevant part of the corresponding Z3py solution generated by our dataset generator:

```python
from z3 import *
subset = [Bool(f'subset_{i}') for i in range(6)]
subset_sum = Sum([If(subset[i], i + 1, 0) for i in range(6)])
constraint_1 = And([Not(And(subset[i],  subset[i + 1])) for i in range(6 - 1)])
constraint_2 = And([Not(And([subset[i], subset[j], subset[k]])) \
                    for i in range(6) for j in range(i + 1, 6) \
                    for k in range(j + 1, 6) if j - i == k - j])
constraint_3 = subset_sum % 10 == 0
constraint_4 = Not(constraint_3)
constraint_5 = And(constraint_2, constraint_4)
constraint_6 = And(constraint_1, constraint_5)
solver = Solver()
solver.add(constraint_6)
```

## 3.2 Rephrasing Problems

We found that fine-tuning on raw synthetic problems leads to severe overfitting to the specific format of those problems (almost perfect scores on our validation synthetic problems), while degrading the performance on real problems from the NuminaMath dataset [8]. To address this issue and make the problem statements sound more natural, we rephrase the problems using a language model.

To minimize the number of incorrect rephrasings, we use a technique similar to rejection sampling. For each problem, we prompt the language model to generate a naturally sounding problem statement. We then prompt it again to generate a new Z3py solution from the rephrased statement. We verify that the original and generated solutions produce the same integer answer. If they do, we count both the rephrasing and the new solution as correct.

In our experiments, we rephrased each problem twice and sampled three solutions for each rephrasing at temperature $0.5$. For example, for the problem above, we generated the following rephrasing.

```
Consider the set {1, 2, 3, 4, 5, 6}.  How many subsets can be formed such that
no two elements in the subset are consecutive integers, no three elements form an
arithmetic progression, and the sum of the elements is not divisible by 10?
```

We constructed training sets containing 2624 and 2883 rephrased synthetic problems, for GPT-4o-mini and GPT-4o, respectively. We generated rephrasings and solutions for GPT-4o-mini and GPT-4o independently to avoid confounding our results with knowledge transfer between the models.

## 3.3 NuminaMath Dataset

We applied our approach to problems from the NuminaMath dataset [8]. We excluded the synthetic-math portion, as it contains rephrasings (which may be incorrect) of other problems. We retained only problems with integer answers and used the GPT-4o-mini model to filter for counting problems (which are potentially solvable using our Z3py approach). For validation, we selected the amc-aime portion of the dataset, resulting in $145$ problems after filtering. The remaining $8935$ filtered problems were used to construct a baseline training set.

For each problem in the training set, we generated 3 Z3py solutions for rejection sampling, using GPT-4o-mini and GPT-4o, respectively. This process left us with $2624$ training problems for GPT-4o-mini and $2883$ training problems for GPT-4o.

## 4 Evaluation

We fine-tuned both GPT-4o and GPT-4o-mini for one epoch using a standard learning rate on three datasets: (i) Problems filtered from the NuminaMath dataset, (ii) synthetic problems, and (iii) a mix of NuminaMath and synthetic problems.

**Z3 solutions**   We evaluated a total of ten models: four before fine-tuning and six after fine-tuning. To assess their performance, we generated 32 samples with a temperature of $0.7$ for each of the $145$ problems in the validation set. We ran the default and fine-tuned versions of GPT4o-mini and GPT4o with a few-shot prompt specifically asking for solutions written in Z3py. We include two example problems with solutions in all prompts. We additionally ran the default models with a Z3py reference prepended to the prompt. A solution was considered correct if it executed properly using Z3py and returned the correct answer. We first compare the models in the pass@k metric [9]. For each $k$, ranging from 1 to 32, we calculated the expected number of problems solved at least once given that we subsample the results on each problem using $k$ samples. For each problem solved $x$ times out of 32 trials, the probability of at least one success in $k$ attempts is given by:

$$P(\text{solved}) = 1 - \frac{\binom{32-x}{k}}{\binom{32}{k}}$$

Summing the probabilities across problems provides the expected number of solved problems per $k$.

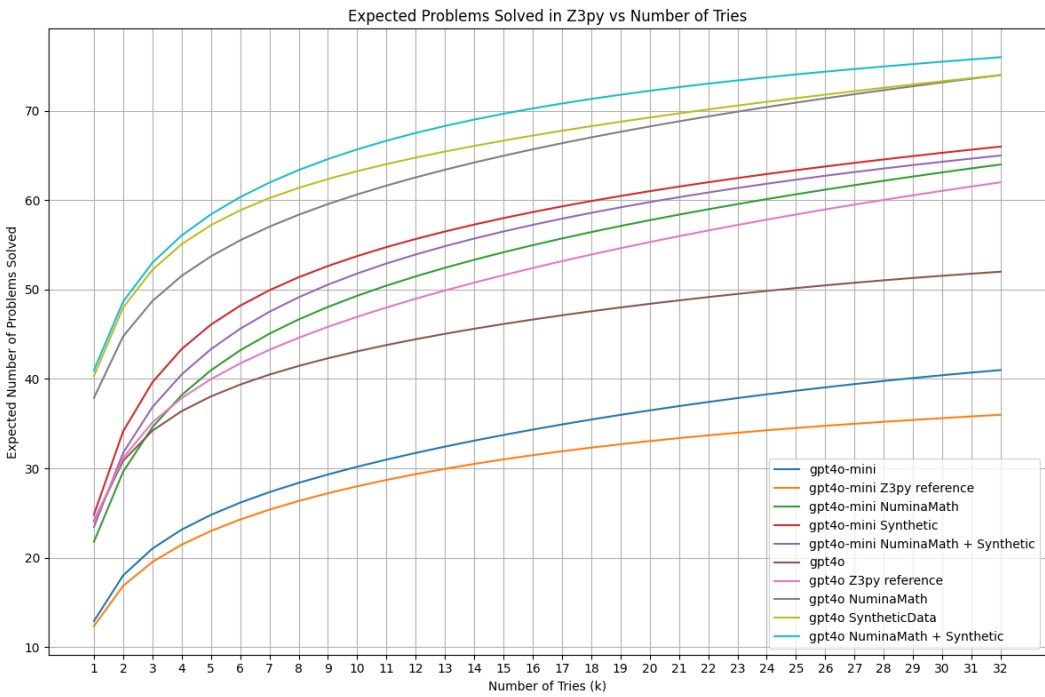

We note that for GPT-4o, fine-tuning on synthetic data outperforms fine-tuning on the NuminaMath dataset, and fine-tuning on a mix of the two outperforms both. For GPT-4o-mini, the results are less clear, as fine-tuning on synthetic data seems to produce the best performance. We also observe that GPT-4o-mini fine-tuned on synthetic data performs better than GPT-4o.

| Model | Score (maj@32) |
|---|---|
| gpt4o-mini | 29 |
| gpt4o-mini Z3py reference | 23 |
| gpt4o-mini NuminaMath | 45 |
| gpt4o-mini Synthetic | 49 |
| gpt4o-mini NuminaMath + Synthetic | 44 |
| gpt4o | 45 |
| gpt4o Z3py reference | 47 |
| gpt4o NuminaMath | 59 |
| gpt4o SyntheticData | 59 |
| gpt4o NuminaMath + Synthetic | 62 |

To enable a fair comparison in a setting where the answers are unavailable, we also calculate the number of correctly solved problems for each model using the majority vote across the 32 samples.

The relative performance of the models aligns with the pass@k metric, supporting the intuition that a model which solves a problem correctly more frequently is more likely to have a majority of correct samples.

**General problem solving**   Additionally, we prompted the models to generate any Python code, potentially using SymPy, Z3Py, or any other library. On our validation data set, all fine-tuned models managed to solve additional tasks that were not solved without our fine-tuning: The strongest difference is on GPT-4o, where fine-tuning with NuminaMath yielded 7 additional solutions on our validation set. Of those solutions, 5 used Z3Py. For GPT-4o-mini, NuminaMath and Numina-Math+Synthetic both yielded 4 extra problems solved, where all of the solutions used Z3Py.

## 5   Conclusion and Future Work

We presented an algorithm for synthesizing training data for fine-tuning LLMs to solve mathematical problem statements using SMT solvers. Our experimental results demonstrate that fully synthetic random problem statements can be helpful for tasks with less readily available training data. Furthermore, we showed that fine-tuning on SMT-based solutions improves the general problem solving capabilities of LLMs. Future work includes more advanced synthetic problem statements, using fine-tuned models to bootstrap harder datasets from existing problem statements, as well as applying the approach to other less well-known libraries for mathematical reasoning in addition to Z3Py.

**Acknowledgments**

This research was partially funded by the Ministry of Education and Science of Bulgaria (support for INSAIT, part of the Bulgarian National Roadmap for Research Infrastructure).

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

# A  Appendix: Prompts Used in Research

## Rephrasing for Synthetic Problems

```
Rewrite the following problem as if it were a question in a middle school math competition.
Ensure that the rewritten question is equivalent to the original while sounding natural.
First, analyze the constraints in the initial problem. Be very careful,
some constraints can contain double negatives.
Provide the final statement formatted as follows: ‘‘‘statement‘‘‘.
```

## Z3 Solver (Fine-Tuning & Evaluation)

```
You will be provided with a combinatorial problem related to enumeration.
Your task is to write Python code using the Z3 library to solve the problem
and print the correct answer.

Requirements:
1. Z3 Solver: Use the Z3 library to model the problem as a constraint satisfaction problem.
2. Correctness: Ensure that your code accurately represents all the constraints of the problem.
3. Efficiency: Structure your code to efficiently find and count all valid solutions.
4. Output: The program should print only the integer answer -- no additional text or
    formatting should be included in the output.

Guidelines:
1. Constraints: Define all necessary variables and constraints clearly.
2. Problem Breakdown: Before writing the code, break down the problem into smaller
components and identify the constraints for each component.
3. Symmetry Breaking: Implement symmetry-breaking techniques where necessary to avoid
    counting equivalent solutions multiple times.
4. Commenting: Add comments to your code to explain the logic and purpose of each part.
```

## Problem Filtering

```
You are given the following problem:

\n\n{problem} \n\n

Determine if it is an enumeration problem. By "enumeration," we mean a complete,
ordered listing of all the items in a collection.
More precisely, the problem must satisfy each of the following conditions:

1. The problem deals with a clearly and explicitly defined,
    finite set of objects (like permutations, combinations, bounded sequences, graphs, etc.).
    Problems involving real numbers, functions, infinite sets, etc.,
    are not considered enumeration problems.
2. There must be clear and explicit constraints on individual objects,
    expressible in first-order logic, that define a subset of the ground set.
    All problems mentioning constraints on subsets of the ground set must be discarded.
3. The main and only objective of the problem must be
    to count the number of elements in the subset defined by the constraints.
    All problems mentioning minimization or maximization must be discarded.
4. A natural solution must involve going through all the elements of the ground set
    and checking if they meet the constraints.
5. The problem must be challenging to solve analytically; i.e.,
    it should not be possible to solve it by a simple formula or direct calculation.
6. The problem statement must not contain any URLs or diagrams.

Do not try to solve the problem. Carefully review each of the listed conditions.
All words in the conditions are important.
```

Please provide a clear answer by putting "Yes" (without quotes) in \boxed{}
if the problem satisfies the criteria, and "No" in \boxed{} otherwise.
If you are not sure, answer "No."

**Example Problem 1**

∗∗Problem∗∗:
Consider the following undirected graph G with 5 vertices:
• Vertices: V = \{1, 2, 3, 4, 5\}
• Edges:    E = \{(1,2), (1,3), (2,3), (3,4), (4,5)\}

We need to assign colors to the vertices using colors A, B, and C, such that:
1. Adjacent vertices must have different colors.
2. Vertex 1 must be colored with color A.
3. The number of vertices colored with color B must be exactly 2.

How many valid colorings of the graph satisfy these constraints?

∗∗Solution∗∗:
Step 1: Define Variables:
  • For each vertex, define an integer variable representing its color.
  • Colors are encoded as integers: A = 0, B = 1, C = 2.
Step 2: Specify Constraints:
  1. Color Assignment Constraints:
    • Each vertex's color variable can take on values 0, 1, or 2.
  2. Adjacency Constraints:
    • For each edge (u, v), ensure that the colors of  u  and  v  are different.
    • Specific Vertex Color Constraint:
    • Vertex 1 must be colored with color A (0).
    • Cardinality Constraint:
    • The number of vertices colored with color B (1) must be exactly 2.
Step 3: Model Counting:
    • Use Z3 to find all possible assignments that satisfy the constraints.
    • Count the number of valid colorings.

∗∗Python Code Using Z3∗∗
'''python
from z3 import ∗

# Create a solver instance
solver = Solver()

# Define colors
colors = {'A': 0, 'B': 1, 'C': 2}

# Define variables for each vertex
v1 = Int('v1')
v2 = Int('v2')
v3 = Int('v3')
v4 = Int('v4')
v5 = Int('v5')

vertices = [v1, v2, v3, v4, v5]

# Each variable can be 0 (A), 1 (B), or 2 (C)
for v in vertices:

```
    solver.add(Or(v == 0, v == 1, v == 2))

# Adjacent vertices must have different colors
edges = [(v1, v2), (v1, v3), (v2, v3), (v3, v4), (v4, v5)]
for (u, v) in edges:
    solver.add(u != v)

# Vertex 1 must be colored with color A (0)
solver.add(v1 == colors['A'])

# The number of vertices colored with color B (1) must be exactly 2
num_B = Sum([If(v == colors['B'], 1, 0) for v in vertices])
solver.add(num_B == 2)

# Function to generate all solutions efficiently
def all_smt(s, initial_terms):
    def block_term(s, m, t):
        s.add(t != m.eval(t, model_completion=True))
    def fix_term(s, m, t):
        s.add(t == m.eval(t, model_completion=True))
    def all_smt_rec(terms):
        if sat == s.check():
            m = s.model()
            yield m
            for i in range(len(terms)):
                s.push()
                block_term(s, m, terms[i])
                for j in range(i):
                    fix_term(s, m, terms[j])
                yield from all_smt_rec(terms[i:])
                s.pop()
    yield from all_smt_rec(list(initial_terms))

# Perform model counting
total_colorings = 0
for _ in all_smt(solver, vertices):
    total_colorings += 1

print(total_colorings)
```

**Example Problem 2**

```
**Problem**
Consider seating 5 people around a circular table. Two of these people are identical twins
and cannot be distinguished from each other.
The other three individuals are Alice, Bob, and Charlie.
Arrangements that can be obtained from each other by rotation of the table are considered identical
 (i.e., seating arrangements are counted up to rotational symmetry).
How many distinct seating arrangements are possible under these conditions?

**Solution**
Step 1: Break Rotational Symmetry
```
   • Since rotating the table doesn't create a new arrangement,
      we can fix one person's seat to eliminate this symmetry. We'll fix Alice at seat 0.
   • Seats: 0 (Alice), 1, 2, 3, 4
   • Remaining Guests: Bob, Charlie, Twin 1, Twin 2

```
Step 2: Define Variables
```

- We need variables to represent who is sitting in each seat (excluding the fixed seat 0).
- Create integer variables for each seat (from seats 1 to 4).
- Each variable will represent the guest sitting in that seat.

Step 3: Encode the Guests
- Since the twins are identical, we'll represent them with the same identifier.
- Guest Identifiers:
- Bob: 1
- Charlie: 2
- Twin (both twins share this identifier): 3

Step 4: Add Constraints
1. Domain Constraints:
- Each seat variable can take values from the set {1, 2, 3}, corresponding to Bob, Charlie, and Twin.
2. Uniqueness Constraints:
- Bob and Charlie must occupy one seat each, while the two twins are indistinguishable but must occupy exactly two seats.
- Ensure that Bob and Charlie are each seated exactly once.
- Ensure that exactly two seats are occupied by the twins.

Step 5: Model Counting
Use Z3 to find all possible seating arrangements that satisfy the constraints.
Count the number of valid seating arrangements.

**Python Code Using Z3**
```python
from z3 import *

# Create a solver instance
solver = Solver()

# Define guest identifiers
GUESTS = {'Bob': 1, 'Charlie': 2, 'Twin': 3}

# Fixed seating: Alice is at seat 0
# Variables for seats 1 to 4
seat_vars = [Int(f'seat_{i}') for i in range(1, 5)]

# Domain constraints: Each seat must have a guest (Bob, Charlie, or Twin)
for var in seat_vars:
    solver.add(Or([var == GUESTS[guest] for guest in GUESTS]))

# Uniqueness constraints for Bob and Charlie
solver.add(Sum([If(var == GUESTS['Bob'], 1, 0) for var in seat_vars]) == 1)
solver.add(Sum([If(var == GUESTS['Charlie'], 1, 0) for var in seat_vars]) == 1)

# Total number of twins must be 2
solver.add(Sum([If(var == GUESTS['Twin'], 1, 0) for var in seat_vars]) == 2)

# Function to generate all seating arrangements
def all_smt(s, initial_terms):
    def block_term(s, m, t):
        s.add(t != m.eval(t, model_completion=True))
    def fix_term(s, m, t):
        s.add(t == m.eval(t, model_completion=True))
    def all_smt_rec(terms):
        if sat == s.check():
            m = s.model()
```

```
            yield m
            for i in range(len(terms)):
                s.push()
                block_term(s, m, terms[i])
                for j in range(i):
                    fix_term(s, m, terms[j])
                yield from all_smt_rec(terms[i:])
                s.pop()
        yield from all_smt_rec(list(initial_terms))

# Perform model counting
total_colorings = 0
for _ in all_smt(solver, seat_vars):
    total_colorings += 1

print(total_colorings)
'''
```

# B    Appendix: Example of a Problem Solved After Fine-Tuning

**Problem Statement**

Find the number of functions $f(x)$ from $\{1, 2, 3, 4, 5\}$ to $\{1, 2, 3, 4, 5\}$
that satisfy $f(f(x)) = f(f(f(x)))$ for all $x$ in $\{1, 2, 3, 4, 5\}$.

**Generated Solution**

To solve this problem using the Z3 library, we need to model the constraints on the function $f$
such that $f(f(x)) = f(f(f(x)))$ for all $x$ in $\{1, 2, 3, 4, 5\}$.
Here's how you can implement this:

```python
from z3 import *

# Create a solver instance
solver = Solver()

# Define the domain and range
domain = range(1, 6)

# Define a function f: {1, 2, 3, 4, 5} -> {1, 2, 3, 4, 5}
f = Function('f', IntSort(), IntSort())

# Add constraints that f maps elements to the domain {1, 2, 3, 4, 5}
for x in domain:
    solver.add(And(f(x) >= 1, f(x) <= 5))

# Add the constraint f(f(x)) = f(f(f(x))) for all x
for x in domain:
    solver.add(f(f(x)) == f(f(f(x))))

# Function to generate all solutions
def all_smt(s, initial_terms):
    def block_term(s, m, t):
        s.add(t != m.eval(t, model_completion=True))
    def fix_term(s, m, t):
        s.add(t == m.eval(t, model_completion=True))
    def all_smt_rec(terms):
        if sat == s.check():
```

```python
            m = s.model()
            yield m
            for i in range(len(terms)):
                s.push()
                block_term(s, m, terms[i])
                for j in range(i):
                    fix_term(s, m, terms[j])
                yield from all_smt_rec(terms[i:])
                s.pop()
    yield from all_smt_rec(list(initial_terms))

# Perform model counting
total_functions = 0
for _ in all_smt(solver, [f(x) for x in domain]):
    total_functions += 1

print(total_functions)
'''
```

