# OpenReview forum: "Constraint-Based Synthetic Data Generation for LLM Mathematical Reasoning"
_NeurIPS.cc/2024/Workshop/MATH-AI — MATH-AI 24_

### Official Review · Reviewer_9S9U · 2024-09-26
**Clear and relevant paper, perhaps in need of some additional experiments or metrics**

**Rating:** 8
**Confidence:** 4

**Review:**

This paper seeks to improve pre-trained LLMs' abilities to use a SMT solver (Z3) to solve simple combinatorics problems.
The primary contribution is a synthetically generated dataset of combinatorics problems paired with Z3 solutions.
Importantly, the authors correctly evaluate on a held-out set of real problem instances; the synthetic problems are only used to train on.

Pros:
- The main part of the paper is well written. The description of the domain and the proposed solution is clear.
- The results are somewhat interesting: they do suggest that even simple ways of generating synthetic Z3 instances can help the models a fair bit.
- The paper is relevant to the topic of the workshop, as SMT solvers are used throughout formal mathematics and automated theorem proving, and recent work has shown that interfacing with them is crucial when using LLMs to automate math.

Cons:
- I'm missing a related work section; I understand this is a short paper and that there is not a lot of space to work with but I would appreciate an appendix being added to compare this work to other prior and contemporary research.
- I'm not convinced pass@k is a meaningful metric here; it's used in code generation because you can tell when you get a correct program (because it passes the unit tests). Thus obtaining 1 correct program in 20 samples is still meaningful. In this domain, you can't tell which of your 20 samples is the correct one (without cheating), so it's not necessarily a meaningful metric.
- I would have liked to see more interpretation of the results. For example, looking at the figure it appears to me that the primary benefit is improved (semantic) diversity in the outputs, particularly when comparing the different finetuning methods (e.g., gpt4o syntheticdata with vs. without numinamath is identical at pass@1 but several % different at pass@20). Maybe quantifying/measuring this in some way would reveal more interesting insights.
- I personally find the use of the words "novel algorithm" a bit too strong to describe a simple way to synthetically generate data from a language for which you have a grammar.
- I would have liked to see some analysis of what parts of your synthetic data it was that actually helps the models - e.g. is there some part of the Z3 grammar that has poor coverage in real data, but your synthetic data covers well?

Overall, I think this paper is a good candidate for the workshop. I look forward to reading the paper again once it has been fleshed out a bit more.

---

### Official Review · Reviewer_NpKp · 2024-10-06
**Reviewer of "LLM Training Data Synthesis for More Effective Problem Solving using Satisfiability Modulo Theories"**

**Rating:** 8
**Confidence:** 4

**Review:**

This paper investigates the intersection of mathematical reasoning with Large Language Models (LLMs), particularly focusing on the use of Z3Py, to solve counting problems. The authors chose Z3Py which excels at counting problems with a 25 small enough answer. This eliminates possible concerns and answer extraction issues that exist for math reasoning datasets such as MATH, Algebra.  A novel and interesting approach for generating synthetic training data designed to fine-tune LLMs is presented, improving their mathematical problem-solving capabilities.

---

### Official Review · Reviewer_NGF8 · 2024-10-07
**Practical method for equipping LLMs with counting abilities**

**Rating:** 7
**Confidence:** 3

**Review:**

**Summary**

This paper explores the use of an SMT solver, Z3Py, in order to address the shortcomings of LLMs on counting problems (e.g., "how many ways can a chessboard be tiled with dominoes?"). Models are asked for python code which utilizes Z3Py in order to output the answer to a particular counting problem. The authors introduce a method of synthetic data generation -- problems with Z3Py-based solutions -- over a reasonably broad collection of counting problems (based on sequences, sets, etc.) They evaluate three copies of gpt4o and gpt4o-mini on the aime-amc subset of the NuminaMath dataset: One finetuned on just (a curated subset of) NuminaMath, one just with synthetic data generated from NuminaMath, and one with a mix of both. Fine-tuning on synthetic (or a mix) produces the best performance. When prompts are relaxed to not mandate Z3Py solutions, the fine-tuned models still outperform the base models and even provide new solutions not using Z3Py.

**Pros**
* Writing is easy to follow
* Straightforward and natural idea
* Presents an interesting and practical method of equipping LLMs with computational abilities
* Demonstrates that this method is successful (with these models on this data)
* The rephrasing idea is interesting

**Cons**
* 102: "We found that..." Worth including the details of this experiment.
* 116: "We generated..." More specifics on this should be included.
* Missing related work -- What else has been done in this area?
* Limited to one dataset and gpt.
* NeurIPS Checklist?

---

### Official Review · Reviewer_KavT · 2024-10-07

**Rating:** 6
**Confidence:** 4

**Review:**

This paper proposes an algorithm for synthesizing counting problems that require Z3py to solve.
With the synthesized data, they fine-tune GPT-4o and GPT-4o-mini to use Z3py.
The fine-tuned models are both better at solving problems using only Z3py and using a series of tools like sympy, Z3py and others.

**Weakness:** Novelty is limited. While People haven't done that for Z3py, this kind of data synthesis is common and used for many other tools/libraries.

**Strength:** The performance is good, and SMT seems to provide a better guarantee for the data synthesis quality. I'm curious how this can be extended beyond counting problems.

---

### Decision · Program_Chairs · 2024-10-08

**Decision:**

Accept

**Comment:**

All reviewers are in support of this paper, and it will make a great contribution to our workshop.